# Integrated Assessment of Heart, Lung and Lower Extremity Veins Using Hand-Held Ultrasound Device in COVID-19 Patients: Feasibility and Clinical Application

**DOI:** 10.3390/diagnostics13040724

**Published:** 2023-02-14

**Authors:** Dominika Filipiak-Strzecka, Jarosław D. Kasprzak, Piotr Lipiec

**Affiliations:** Department of Cardiology, Medical University of Lodz, Bieganski Hospital, Kniaziewicza 1/5, 91-347 Lodz, Poland

**Keywords:** hand-held ultrasound devices, COVID-19, lung ultrasound

## Abstract

The emergence of the COVID-19 pandemic caused a significant shortage of medical personnel and the prioritization of life-saving procedures on internal medicine and cardiology wards. Thus, the cost- and time-effectiveness of each procedure proved vital. Implementing elements of imaging diagnostics into the physical examination of COVID-19 patients could prove beneficial to the treatment process, providing important clinical data at the moment of admission. Sixty-three patients with positive COVID-19 test results were enrolled into our study and underwent physical examination expanded with a handheld ultrasound device (HUD)—performed bedside assessment included: right ventricle measurement, visual and automated LVEF assessment, four-point compression ultrasound test (CUS) of lower extremities and lung ultrasound. Routine testing consisting of computed-tomography chest scanning, CT—pulmonary angiogram and full echocardiography performed on a high-end stationary device was completed in the following 24 h. Lung abnormalities characteristic for COVID-19 were detected in CT in 53 (84%) patients. The sensitivity and specificity of bedside HUD examination for detecting lung pathologies was 0.92 and 0.90, respectively. Increased number of B-lines had a sensitivity of 0.81, specificity 0.83 for the ground glass symptom in CT examination (AUC 0.82; *p* < 0.0001); pleural thickening sensitivity 0.95, specificity 0.88 (AUC 0.91, *p* < 0.0001); lung consolidations sensitivity 0.71, specificity 0.86 (AUC 0.79, *p* < 0.0001). In 20 patients (32%), pulmonary embolism was confirmed. RV was dilated in HUD examination in 27 patients (43%), CUS was positive in two patients. During HUD examination, software-derived LV function analysis failed to measure LVEF in 29 (46%) cases. HUD proved its potential as the first-line modality for the collection of heart–lung–vein imaging information among patients with severe COVID-19. HUD-derived diagnosis was especially effective for the initial assessment of lung involvement. Expectedly, in this group of patients with high prevalence of severe pneumonia, HUD-diagnosed RV enlargement had moderate predictive value and the option to simultaneously detect lower limb venous thrombosis was clinically attractive. Although most of the LV images were suitable for the visual assessment of LVEF, an AI-enhanced software algorithm failed in almost 50% of the study population.

## 1. Introduction

More than two years ago, a novel coronavirus (SARS-CoV-2) emerged to pose a major and oftentimes overwhelmingly difficult challenge for medical professionals worldwide. The scale of the problem appeared to be unprecedented, demanding the safe delivery of intensive healthcare for hundreds of thousands of patients by healthcare workers [1]. Most healthcare systems faced the problem of overload including limited access to diagnostic procedures due to medical staff and equipment scarcity and elevated personnel protection standards [2,3,4]. In this setting novel alternative diagnostic alternatives gained interest, including point-of-care ultrasound (POCUS) protocols, which are focused on bedside, real-time diagnostic data acquisition with limited medical personnel exposure [5,6]. Evidence supporting the implementation of POCUS has been steadily increasing [7,8]. Certain improvements in the technical aspects of equipment development led to the emergence and the current position of handheld ultrasound devices (HUD). They have gained a particular recognition due to their extremely simplified user interface and capability of wireless, real-time quantification analysis of captured images. Even though available imaging modes are usually limited to grayscale two-dimensional (2D) imaging, color Doppler and simplified measurements, their extended use during the pandemic appeared promising, as the transportation of a conventional high-end stationary device was highly impractical due to size, risk of damage or cross-infection. On the other hand, the amount of clinical data provided by such a device is oftentimes excessive during the initial assessment and easiness of operation is designed for an examiner with significant skillset and experience. During the very intensive, fast-paced shifts at the COVID ward, looking after an expensive device rather than focusing all the effort on patients’ well-being could be unnecessarily distractive. Smartphone- or tablet-sized devices are much better suited for effortless transportation and point-of-care patient evaluation. COVID-19 symptoms are known to be centered around the respiratory system. Although the findings on lung ultrasonography (LUS) during COVID-19 infection appear to be nonspecific and are likely comparable with similar non-COVID respiratory diseases, comprehensive ultrasound imaging of the lungs and surrounding structures may improve clinical decision processes in patients with COVID-19 and comorbidities [9,10]. Very good correlation of LUS findings with chest computed tomography results has been previously confirmed [11,12,13], and in some scenarios LUS can be considered an alternative to ionizing radiation imaging. Buonsenso et al. proposed that LUS could replace stethoscopes in the ongoing COVID-19 pandemic [14]. Considering that cardiovascular diseases represent the most severe and frequent COVID-19 complications [15], fast access to echocardiography might be useful; however, the procedure requires prolonged and close contact with patient. Hence, the use of short imaging protocol which reduces acquisition time and minimizes high-risk exposure during the pandemic has been discussed [16,17]. We hypothesized that integrated, limited, bedside assessment of lung, heart and lower extremity veins with the use of AI-enhanced HUD may provide clinically relevant information within minutes.

## 2. Materials and Methods

Sixty-three (41 men, mean age 63 ± 11) consecutively hospitalized COVID-19 ward patients with a confirmed diagnosis of SARS CoV-2 infection were prospectively included in the study population between November 16th and December 23rd 2020. The infection was confirmed by real-time PCR or antigen testing. Two patients were not included due to brief hospitalization time, during which the main researcher/examiner was delegated to pursue other clinical chores.

Signed informed consent was obtained from each patient. The study protocol was approved by the ethics committee of our institution.

Within the first 24 h of hospitalization, initial patient assessment was augmented with bedside ultrasound diagnostics performed with HUD which included the assessment of the heart, lung and large veins of lower extremities. HUD examination was performed by the last-year cardiology resident after 6-months of hands-on practice in the echocardiographic room. The device used in the study was a Vscan Extend (GE Vingmed Ultrasound, Horten, Norway); it featured a dual probe, combining a phased array probe (frequency range of 1.7–3.8 MHz, image sector limited to 70°, maximum depth 24 cm, aperture size 13 × 19 mm) with a linear probe (frequency range of 3.4–8.0 MHz). Vscan Extend enables both the 2D grayscale and Doppler mode. The mean time of bedside HUD examination was 4.4 ± 1.1 min.

### 2.1. HUD—Lung Assessment

In total, fourteen lung areas were examined: paravertebral basal, middle and apical, midaxillary basal and upper and midclavicular basal and upper. In cases of performance of LUS examinations in the setting of the intensive care ward and in patients unable to remain in a seated position, the posterior areas were excluded from evaluation. The presence of an increased number of B-lines (more than 3 in one view), pleural thickening (the increase in thickness of the pleura of more than 3 mm, with pleural line irregularity) and subpleural consolidation (hypoechogenic subpleural structures, assessed visually) were recorded as abnormal. Linear probe and pulmonic preset were used.

### 2.2. HUD—Cardiac Assessment

The protocol included a goal-oriented examination, focused on right ventricular (RV) size and left ventricular global function. RV diameter was measured in the right-ventricle-focused 4 chamber apical view. RV enlargement was diagnosed if the basal diameter exceeded 41 mm. Left ventricle ejection fraction was assessed visually and automatically, with the use of AI-enhanced pre-installed Lvivo App EF software (DiA Imaging Analysis Ltd., Be’er Sheva, Israel). Recording lasting at least two heart cycles of four chamber apical view, with depth adjusted so 2/3 of the view were occupied by the left ventricle, was obtained. The projection acquisition was completed using the device’s standard cardiac preset in accordance with the manufacturer’s guidelines. Calculations such as LVEF, end-systolic volume and end-diastolic volume were made automatically in real time, based on endocardial border tracing. Should the algorithm fail, the recording was repeated, and three consecutive failures were recorded as a case in which automated LVEF assessment was not possible.

### 2.3. HUD—Vein Compression Ultrasound Test

Four-point venous compression ultrasound test (CUS) was conducted with linear probe and vascular preset. Patient was lying down on their back during the examination. The femoral vein was assessed distally from the inguinal ligament to the area located 2 cm distally to the junction of the common femoral vein and the greater saphenous vein. The collapsing of both common and deep femoral veins was evaluated. The popliteal vein was assessed from the level of the popliteal fossa right up to the level of its trifurcation. The direct pressure applied with the transducer was used in order to completely occlude the lumen of the vein. Should the vein lumen be completely occluded, a diagnosis of a DVT at this area was excluded. If the vein lumen could not be occluded, the test result was considered positive.

### 2.4. Standard Imaging

During the next 24 h, all patients underwent a chest computed tomography (CT) scan, computed tomography pulmonary angiogram and full echocardiographic examination performed on a high-end stationary device, the results of which were treated as a reference.

Lung abnormalities in CT and LUS were compared according to the following criteria: the equivalent of ground glass opacities were a presence of 3 or more B-lines in one view and pleural thickening in CT—thickened hyperechoic pleural line in LUS and the presence of subpleural consolidation were assessed in both imaging methods.

### 2.5. Statistical Analysis

Continuous and categorical variables were expressed as mean ± SD and as percentages (%), respectively. Agreement between LVEF derived from standard echocardiography and HUD auto EF measurements was calculated with intraclass correlation coefficient and Bland–Altman analysis. The 95% limits of agreement were defined as the range of values between ± 1.96 standard deviations from the mean difference. ROC curve analysis was performed and the correlation coefficient was calculated to determine the concordance between HUD assessment versus full echocardiographic examination findings and lung ultrasound HUD assessment versus computed tomography. Weighted Cohen’s kappa coefficient was calculated to determine the agreement between the methods—strength of agreement was categorized as follows: 0–0.20 poor, 0.21–0.40 fair, 0.41–0.60 moderate, 0.61–0.80 good and 0.81–1.00 very good. Sensitivity, specificity, positive predictive value (PPV) and negative predictive value (NPV), were calculated using 2 × 2 contingency tables and the corresponding 95% confidence intervals were determined.

## 3. Results

The basic characteristics of the study population are presented in Table 1.

### 3.1. Lung Assessment

Lung lesions typical for COVID-19 were confirmed in CT in 53 (84%) patients (Figure 1). Ground glass opacities were present in 50 patients, pleural thickening in 21 patients and subpleural consolidation in 28 patients. Lung abnormalities were detected in HUD examination in 50 patients (>3 B-lines in 43 patients, thickened pleural line in 25 patients and subpleural consolidation in 25 patients) (Figure 1). The sensitivity and specificity of bedside HUD examination for diagnosing lung involvement was 92% and 90% retrospectively, AUC = 0.92, *p* < 0.0001. Weighted Cohen’s kappa was 0.735 (± 0.111, 95% CI 0.517–0.953). The highest concordance with CT was found for pleural thickening (kappa 0.788 ± 0.07, 95% CI 0.651–0.926). The detailed comparisons of bedside HUD examination and CT for diagnosing lung involvement are presented in Table 2.

### 3.2. Cardiac Assessment

Mean LVEF in standard echocardiography was 46 ± 12%. During HUD examination, automated LV function analysis software failed to calculate LVEF in 29 (46%) cases due to suboptimal image quality (Figure 2). In the remaining cases, the mean value of LVEF was 47 ± 14%. The intraclass correlation coefficient between the LVEF values obtained with LVivo and the reference method was r = 0.84 (*p* < 0.0001, 95 CI 0.70–0.92). In Bland–Altman analysis, the lower and upper limits of agreement were −12.30 and 17.9, respectively. However, it should be mentioned that in individual cases the plot revealed relatively large discrepancies between both methods of LVEF measurements exceeding 1.96 SD (Figure 3). The feasibility of visual LVEF assessment was 84%. The intraclass correlation coefficient of visual assessment and full standard echocardiography was 0.94 (95% CI: 0.91–0.97).

RV was found to be dilated in HUD examination in 27 patients (43%) and in 31 patients (49%) in full echocardiographic examination. Weighted kappa was 0.803 ± 0.074, 95% CI 0.665–0.953.

### 3.3. Vein Compression Ultrasound

The feasibility of compression ultrasound tests performed with the use of HUD was high. In total, 100% of femoral vein and 94% of popliteal veins imaging were classified to be sufficient for reliable assessment. In two patients the CUS result was positive, consistent with further observed clinical course.

### 3.4. Pulmonary Embolism Diagnosis

In 20 patients (32%), pulmonary embolism was confirmed by angioCT—in 10 among them embolism was limited to the subsegmental arteries. RV enlargement treated as a Graph was changed as suggested.marker of PE had low sensitivity and specificity (60% and 65%, respectively), AUC = 0.62 ± 0.067; 95% CI 0.495 to 0.744; *p* = 0.06. Positive predictive value of RV enlargement as a marker of pulmonary embolism was 45% and negative predictive value was 78%. For proximal PE, only the area under the ROC curve increased to 0.721 (±0.074; 95% CI 0.593 to 0.826; *p* = 0.003) sensitivity and specificity were 80% and 64%, respectively, PPV 30%, NPV 94%.

In both cases of positive CUS pulmonary embolism was confirmed in angioCT (Figure 4).

## 4. Discussion

Our study has shown that HUD imaging capabilities, despite inherent device-related limitations, proved sufficient for lung screening in search of the COVID-19 symptoms. Certain findings on LUS correlated well with the corresponding abnormalities detected in the CT lung scan. However, automated LVEF assessment provided by the HUD pre-installed application was not possible in a significant percentage of enrolled patients, and in the remaining cases the quality of imaging was not up to the standard of clinical feasibility. Interestingly, RV enlargement proved to be relatively often detected in COVID-19 patients regardless of the concomitant PE. Presented results concerning the sensitivity and specificity of bedside HUD examination for the detection of lung abnormalities were on-par with the ones obtained in the studies performed with the stationary devices [18].

The clinical feasibility of HUD in diverse clinical scenarios was previously confirmed in numerous studies [19,20,21,22,23,24,25,26,27,28]. In the COVID-19 dominated time, the fundamental advantages of HUD such as ultra-portability and an accessible user-friendly interface prove additionally valuable. Bedside examination enhanced with the elements of imaging diagnostics performed during routine clinical care decreased the risk of cross infection also by avoiding the need to disinfect/sterilize high-end workstations. Limited usage of personal protective equipment is also a considerable benefit [29]. The relatively low price of HUD can also mean that it might become a dedicated, COVID-19-patients-only diagnostic device, which may also limit the risk of viral transmission.

The most common abnormalities encountered in LUS were increased number of B-lines (discrete or confluent, multifocal and usually bilateral), thickening of pleura with pleural line irregularities and small subpleural consolidations (<1 cm height), which progress to large, poorly vascularized or avascular consolidations (>1 cm height) [30]. LUS sensitivity for the detection of lung abnormalities such as pleural thickening, subpleural consolidation, and ground-glass opacification equivalent to CT was confirmed. Both specificity and sensitivity calculated from the results of our study were high. Some limitations of the described method were also noted, e.g., LUS may not be able to detect a centrally located consolidation resulting from a bacterial superinfection. Furthermore, similarly to other diagnostic tools, ultrasound is often unable to differentiate acute versus chronic lesion, limiting its power of early COVID-19 pneumonia diagnosis in the population with preexisting pulmonary conditions [31].

POCUS has demonstrated excellent accuracy in the detection of a DVT across a spectrum of settings and providers [32,33,34]. In our study population, a positive CUS result was a rare finding observed in only one patient. This observation contrasts with the results of other studies suggesting that patients with COVID-19 pneumonia frequently develop thrombotic complications, including deep venous thrombosis [7,35,36,37]. A meta-analysis demonstrated a 20% venous thromboembolism rate in hospitalized patients with COVID-19 [38]. This discrepancy may be due to the limited study population; however, a potentially higher rate of superficial venous thrombi should also be taken into consideration. It was suggested that the six-point scan rather than the standard four-point scan should be performed [7].

Despite the promising results of our previous study inspecting the accuracy of PE diagnosis established with HUD in the pre-COVID-19 era, our current results are somewhat different. Expectedly, RV enlargement was commonly detected within our study population consisting of pneumonia patients. Respiratory acidosis, alveolar inflammatory edema and microvascular alterations may increase pulmonary vascular resistance [39] and positive pressure ventilation may further increase RV afterload [40], which may lead to RV dilatation, also without a concomitant PE [15]. Moreover, in half of the diagnosed PE cases thrombi were located in subsegmental arteries, which may explain the smaller percentage of RV volumetric overload in our study population with PE. In situ thrombosis occurring in smaller pulmonary arteries can also be proposed to explain the lower sensitivity of HUD both for PE diagnosis and venous thrombosis in our study group.

Automated LVEF evaluation is a HUD feature of high clinical relevance. It has been previously reported that 17% of hospitalized COVID-19 cases are complicated with acute cardiac injury (ACI), which has a serious implication regarding mortality rate [40,41,42,43,44]. Visual quantification of left ventricle systolic function typically requires long-term learning and training processes [45,46,47,48] and might be particularly challenging for the non-expert echocardiographers. The capability of LVEF assessment with LVivo software was confirmed in previous research; however, sufficient acoustic window quality was vital for the success of the algorithm [49]. Results of our study—large number of patients with failed automated LVEF assessment, poor agreement between the LVEF values derived from HUD and standard echocardiography in numerous other cases, may suggest that imaging quality was generally not sufficient for this purpose. This may probably be supported by the personal experience of any clinician treating COVID-19 patients dealing with factors such as breathlessness, forced patient’s body position, the need for obtaining projections in a non-standard patient body position (e.g., while lying on the right side) and, finally, worse device handling and screen visibility obscured by the protective equipment. It is worth noting that algorithm used in our study was fully automated without any manual contour edition capabilities. In the more recently introduced HUD, a new pre-installed software allows the examiner to manually trace the endocardial borders. This might address at least some of technical constraints resulting in algorithm failure. The above mentioned functionality further enhanced by the addition of a system guiding the examiner to obtain the correct projections might prove helpful for the less-experienced sonographers.

## 5. Conclusions

HUD proved its potential as the first-line modality for the collection of heart–lung–vein imaging information among patients with severe COVID-19. HUD-derived diagnosis was especially effective for the initial assessment of lung involvement, identifying lung lesions typical for COVID-19 with high sensitivity and specificity.

Expectedly, in this group of patients with a high prevalence of severe pneumonia, HUD-diagnosed RV enlargement had moderate predictive value for pulmonary embolism, and the option to simultaneously detect lower limb venous thrombosis was clinically attractive. Although most of LV images were suitable for the visual assessment and ventricular quantification was usually consistent with deferred full transthoracic echocardiogram, in this population with typically difficult acoustic windows an AI-enhanced software algorithm failed to calculate LV ejection fraction in almost 50% of the study population.

## 6. Limitations

This is a single center study with a limited study population. All examinations were performed by a researcher with a limited echocardiographic experience. One could assume that a more experienced echocardiographer could achieve better results. However, in the wake of a significant shortage of expert care, we strongly believe that the examiner with limited experience facing significant diagnostic challenges is much closer to the clinical reality of the COVID-19 pandemic.

## Figures and Tables

**Figure 1 diagnostics-13-00724-f001:**
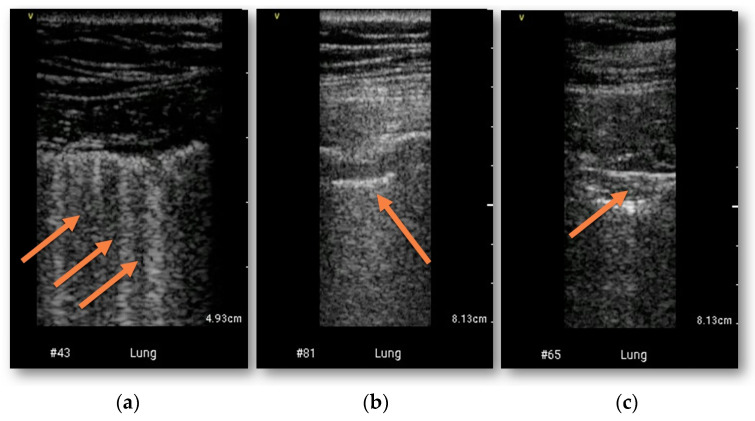
Examples of lung ultrasonography performed with HUD. (**a**) increased number of B-lines; (**b**) pleural thickening; (**c**) subpleural consolidation (abnormalities marked with arrows).

**Figure 2 diagnostics-13-00724-f002:**
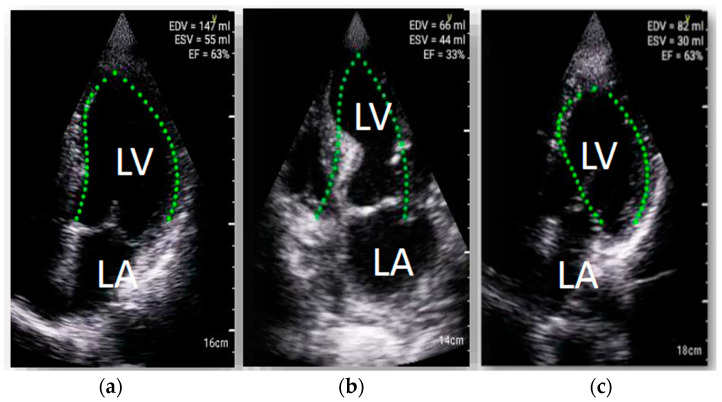
Examples of LVivo software implementation in patients with COVID-19; software failed to calculate LVEF in 29 (46%) cases. Four-chamber apical view, (**a**) successful left ventricle ejection fraction measurements with LVivo software, (**b**,**c**) examples of LVivo software failure in endocardial border detection examples of lung ultrasonography performed with HUD.

**Figure 3 diagnostics-13-00724-f003:**
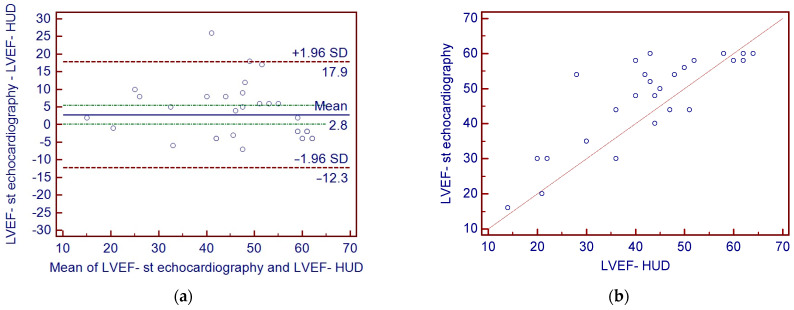
LVivo (HUD) and reference LVEF measurements comparison; (**a**) Bland–Altman plots of the LVEF assessed with LVivo software and reference method; (**b**) scatter diagram correlation.

**Figure 4 diagnostics-13-00724-f004:**
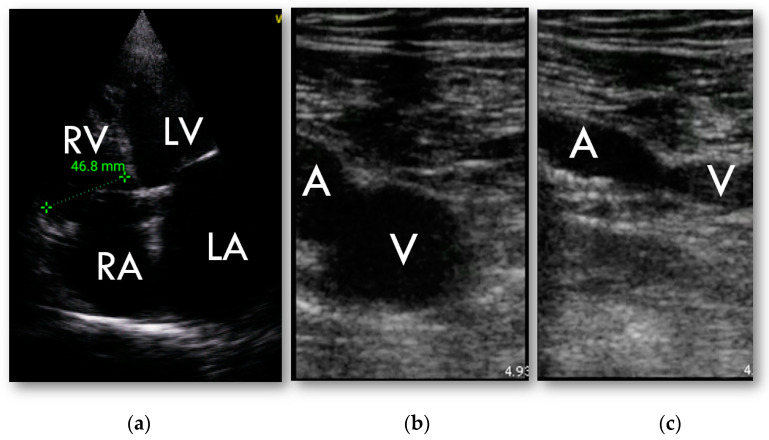
HUD examination; (**a**)—RV focused 4-chamber apical view, basal dimension; (**b**,**c**)—compression ultrasound test of common femoral vein: (**b**)—without compression; (**c**)—abnormal examination in deep vein thrombosis. Vein is not entirely occluded.

**Table 1 diagnostics-13-00724-t001:** Characteristics of study population.

	Number of Patients [Percentage]
Total number of patients	63
Male sex	41 [65]
Mean age (y)	63 ± 11
History of: myocardial infarction	23 [37]
Chronic heart failure	27 [43]
Arterial hypertension	39 [62]
Atrial fibrillation	12 [19]
Diabetes mellitus	19 [30]
Obesity	33 [52]
Chronic kidney disease	21 [33]
Smoking	17 [27]
Initial O_2_% saturation	89 ± 3.3
Mean CRP [mg/mL]	86.3 ± 75.8
Mean NT-proBNP [pg/mL]	2583 ± 3397
Mean D-dimer [ugFEU/L]	1988 ± 2352
Mean TnT [ng/mL]	0.401 ± 0.94

**Table 2 diagnostics-13-00724-t002:** Diagnostic value parameters of lung HUD examination. Chest CT scan was treated as a reference.

	No. of Patients [%]	Sensitivity of HUD Examination	Specificity of HUD Examination	AUC	*p* Value	Weighted Kappa
**Lung involvement**	50 [79%]	92%	90%	0.92	<0.0001	0.735
**Increased No. of B-lines**	43 [68%]	81%	83%	0.82	<0.0001	0.569
**Pleural thickening**	25 [40%]	95%	88%	0.91	<0.0001	0.788
**Lung consolidation**	25 [40%]	71%	86%	0.79	<0.0001	0.593

## Data Availability

The data presented in this study are openly available in FigShare at [https://doi.org/10.6084/m9.figshare.22093250].

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
