# Peer review of "Integrated Assessment of Heart, Lung and Lower Extremity Veins Using Hand-Held Ultrasound Device in COVID-19 Patients: Feasibility and Clinical Application"

_diagnostics, 2023, doi:10.3390/diagnostics13040724_

Round 1

Reviewer 1 Report

sample size is small.

Some modification of English language is required.

Author Response

Dear Sir or Madam,

             We would like to express our gratitude for your comments regarding the submitted manuscript. We found them very interesting, relevant and well-grounded. We endeavored to address these issues as thoroughly as possible. Please find below the cited comments (C) followed by our answers (A) and description of alterations made to the manuscript.

C: Sample size is small.

A: Our study should be treated as a pilot study demonstrating potential capabilities of handheld ultrasound device in an integrated assessment of patients diagnosed COVID-19. Nevertheless, we have added ‘Limitations’ section, in which we have addressed this issue.

Page 10 line 325

 This is the single center study with limited study population. All examinations were performed by the researcher with a limited echocardiographic experience(…)

C: Some modification of English language is required.

A: Paper was revised and errors corrected.

We hope we have addressed your comments thoroughly enough. Should you have any further questions we will be very happy to address them.

Thank you for your time.

On behalf of the authors.

Reviewer 2 Report

The author presented a preliminary experience in the use of HUD in the initial clinical assessment of COVID-19 patients. 

The authors declare that HUD could be useful, especially in the lungs and respiratory assessment while cardiac evaluation is still debatable. 

The abstract must be structured: Introduction, Methods, Results and conclusion.

The Authors have to clarify the design of the study: prospective, retrospective, etc... Moreover they have to specify the time period of study. 

I would like to know also if all patients hospitalized were screened by HUD or some of them were not included. If the answer if yes, please clarify the number and the cause of the exclusion. 

In the methods section, the Authors have to clarify the way they conducted the study. No data must be included in this section, so table I must be moved in the results section. In addition, table I is somekind poor in term of baseline patients' features. Could the Author implement it? 

In terms of outcomes, do you think that the experience of operators could have impacted the reliability of HUD especially in cardiac assessment? This could be a bias. 

Author Response

Dear Sir or Madam,

             We would like to express our gratitude for your comments regarding the submitted manuscript. We found them very interesting, relevant and well-grounded. We endeavored to address these issues as thoroughly as possible. Please find below the cited comments (C) followed by our answers (A) and description of alterations made to the manuscript.

C: The abstract must be structured: Introduction, Methods, Results and conclusion.

A: According to the instructions for authors published on  a journal website the use of headings is not recommended. However, we have split the abstract into paragraphs for the sake of text clarity.

C: The Authors have to clarify the design of the study: prospective, retrospective, etc... Moreover they have to specify the time period of study.

A: The methods section was modified and required information were added:

Page 2 line 78

There was:

63 patients (41 men, mean age 63±11) consecutively hospitalized COVID-19 ward patients with the confirmed diagnosis of SARS COV-2 infection were included into the study population. The infection was confirmed by real-time PCR or antigen testing.

There is:

63 patients (41 men, mean age 63±11) consecutively hospitalized COVID-19 ward patients with the confirmed diagnosis of SARS COV-2 infection were prospectively included into the study population between November 16th and December 23th 2020. The infection was confirmed by real-time PCR or antigen testing.

C: I would like to know also if all patients hospitalized were screened by HUD or some of them were not included. If the answer if yes, please clarify the number and the cause of the exclusion.

A: Study population consists of consecutively hospitalized patients within a mentioned timeframe. 2 patients were not included due to the brief hospitalization time during which the main researcher/examiner was delegated to pursue other clinical chores. (First patient – transferred to the surgical ward with acute pancreatitis; second – severe general condition requiring mechanical ventilation; died <12 h from the admission). The information was added to the methods section.

Page 2 line 82

2 patients were not included due to the brief hospitalization time, during which the main researcher/examiner was delegated to pursue other clinical chores.

C: In the methods section, the Authors have to clarify the way they conducted the study. No data must be included in this section, so table I must be moved in the results section. In addition, table I is somekind poor in term of baseline patients' features. Could the Author implement it?

A: As suggested, tables were moved to the results section. Table 1 was accordingly expanded.

C: In terms of outcomes, do you think that the experience of operators could have impacted the reliability of HUD especially in cardiac assessment? This could be a bias.

A: Sonographer’s experience could have affected the results; but during the COVID -19 pandemic and in the wake of significant shortage of expert care we strongly believe that examiner with limited experience facing significant diagnostic challenges is much closer to the clinical reality of that time. In such circumstances, a brief yet prompt patient’s echocardiographic assessment was most beneficial to the diagnostic process.

However, we have added some additional information to the ‘limitations’ section.

Page 10 line 325

All examinations were performed by the researcher with a limited echocardiographic experience. One could assume that a more experienced echocardiographer could achieve better results. However, in the wake of significant shortage of expert care we strongly believe that examiner with limited experience facing significant diagnostic challenges is much closer to the clinical reality of COVID-19 pandemic.

We hope we have addressed your comments thoroughly enough. Should you have any further questions we will be very happy to address them.

Thank you for your time.

On behalf of the authors.

Reviewer 3 Report

I read with great attention the paper entitled "Integrated assessment of heart, lung and lower extremity veins using hand-held ultrasound device in COVID-19 patients: feasibility and clinical application"

I would congratulate the Authors for their well reported preliminary clinical experience in COVID Patients.

I have some minor suggestions:

- Author should better explain the differences and the advantages (if any) of the Handheld Ultrasound Device compared with conventional devices

- Author should clarify why a Resident performed the assessment instead of a more expert Physician, and if this decision could have affected study results

- A 46% failure rate was reported in software-derived LV function analysis, this aspect should be better explained and discussed .

Author Response

Dear Sir or Madam,

             We would like to express our gratitude for your comments regarding the submitted manuscript. We found them very interesting, relevant and well-grounded. We endeavored to address these issues as thoroughly as possible. Please find below the cited comments (C) followed by our answers (A) and description of alterations made to the manuscript.

C: Author should better explain the differences and the advantages (if any) of the Handheld Ultrasound Device compared with conventional devices

A: Following fragment was added to the introduction section.

Page 2 line 54

There was:

Certain improvements in the technical aspects of equipment development led to the emergence and the current position of the handheld ultrasound devices (HUD). They have gained a particular recognition due to their extremely simplified user interface and capability of wireless, real-time quantification analysis of captured images. Even though available imaging modes are usually limited to greyscale two-dimensional (2D) imaging, color Doppler and simplified measurements, their extended use during the pandemic appeared promising.

There is:

Certain improvements in the technical aspects of equipment development led to the emergence and the current position of the handheld ultrasound devices (HUD). They have gained a particular recognition due to their extremely simplified user interface and capability of wireless, real-time quantification analysis of captured images. Even though available imaging modes are usually limited to greyscale two-dimensional (2D) imaging, color Doppler and simplified measurements, their extended use during the pandemic appeared promising, as the transportation of a conventional   high-end stationary device was highly impractical due to the size, risk of damage or cross-infection. On the other hand the amount of clinical data provided by such device is oftentimes excessive during the initial assessment and easiness of operation is designed for the examiner with significant skillset and experience. During the very intensive, fast-paced shifts at the Covid ward, looking after an expensive device rather than focusing all the effort on patients well-being could be unnecessarily distractive. Smartphone- or tablet-sized device is much better suited for effortless transportation and point-of-care patient evaluation.

C: Author should clarify why a Resident performed the assessment instead of a more expert Physician, and if this decision could have affected study results

A: Sonographer’s experience could have affected the results; but during the COVID -19 pandemic and in the wake of significant shortage of expert care we strongly believe that examiner with limited experience facing significant diagnostic challenges is much closer to the clinical reality of that time. In such circumstances, a brief yet prompt patient’s echocardiographic assessment was most beneficial to the diagnostic process.

However, we have added some additional information to the ‘limitations’ section.

Page 10 line 325

All examinations were performed by the researcher with a limited echocardiographic experience. One could assume that a more experienced echocardiographer could achieve better results. However, in the wake of significant shortage of expert care we strongly believe that examiner with limited experience facing significant diagnostic challenges is much closer to the clinical reality of COVID-19 pandemic.

C: A 46% failure rate was reported in software-derived LV function analysis, this aspect should be better explained and discussed.

A: Related discussion paragraph was expanded.

Page 9 line 289

There was:

Automated LVEF evaluation is a HUD feature of high clinical relevance. It has been previously reported, that 17% of hospitalized COVID-19 cases is complicated with acute cardiac injury (ACI) which has a serious implication regarding mortality rate. Visual quantification of left ventricle systolic function typically requires long-term learning and training process and might be particularly challenging for the non-expert echocardiographers. Capability of LVEF assessment with LVivo software was confirmed in previous research, however, sufficient acoustic window quality was vital for the success of the algorithm. Results of our study - large number of patients with failed automated LVEF assessment, poor agreement between the LVEF values derived from HUD and standard echocardiography in numerous other cases may suggest that imaging quality was generally not sufficient for this purpose. This may probably be supported by the personal experience of any clinician treating COVID-19 patients dealing with factors such as breathlessness, forced patient’s body position, the need for obtaining projections in a non-standard patient body position (e.g. while lying on the right side) and, finally, worse device handling and screen visibility obscured the protective equipment.

There is:

Automated LVEF evaluation is a HUD feature of high clinical relevance. It has been previously reported, that 17% of hospitalized COVID-19 cases is complicated with acute cardiac injury (ACI) which has a serious implication regarding mortality rate. Visual quantification of left ventricle systolic function typically requires long-term learning and training process and might be particularly challenging for the non-expert echocardiographers. Capability of LVEF assessment with LVivo software was confirmed in previous research, however, sufficient acoustic window quality was vital for the success of the algorithm. Results of our study - large number of patients with failed automated LVEF assessment, poor agreement between the LVEF values derived from HUD and standard echocardiography in numerous other cases may suggest that imaging quality was generally not sufficient for this purpose. This may probably be supported by the personal experience of any clinician treating COVID-19 patients dealing with factors such as breathlessness, forced patient’s body position, the need for obtaining projections in a non-standard patient body position (e.g. while lying on the right side) and, finally, worse device handling and screen visibility obscured the protective equipment.  It is worth noting that algorithm used in our study was fully automated without any manual contour edition capabilities. In the more recently introduced HUD, a new pre-installed software allows the examiner to manually trace the endocardial borders. This might address at least dome of technical constraints resulting in algorithm failure. The above mentioned functionality further enhanced by the addition of a system guiding the examiner to obtain the correct projections which might prove helpful for the less-experienced sonographers.

We hope we have addressed your comments thoroughly enough. Should you have any further questions we will be very happy to address them.

Thank you for your time.

On behalf of the authors.

Round 2

Reviewer 2 Report

The paper has increased its quality after revision.